# Selective control of synaptic plasticity in heterogeneous networks through transcranial alternating current stimulation (tACS)

**Aref Pariz**[1,2]*, **Daniel Trotter**[2,3], **Axel Hutt**[4], **Jeremie Lefebvre**[1,2,3,5]

**1** Department of Biology, University of Ottawa, Ontario, Canada, **2** Krembil Brain Institute, University Health Network, Toronto, Ontario, Canada, **3** Department of Physics, University of Ottawa, Ontario, Canada, **4** Team MIMESIS, INRIA - UMR7357 CNRS - ICube, Strasbourg, France, **5** Department of Mathematics, University of Toronto, Ontario, Canada

* apariz@uottawa.ca

## Abstract

Transcranial alternating current stimulation (tACS) represents a promising non-invasive treatment for an increasingly wide range of neurological and neuropsychiatric disorders. The ability to use periodically oscillating electric fields to non-invasively engage neural dynamics opens up the possibility of recruiting synaptic plasticity and to modulate brain function. However, despite consistent reports about tACS clinical effectiveness, strong state-dependence combined with the ubiquitous heterogeneity of cortical networks collectively results in high outcome variability. Introducing variations in intrinsic neuronal timescales, we explored how such heterogeneity influences stimulation-induced change in synaptic connectivity. We examined how spike timing dependent plasticity, at the level of cells, intra- and inter-laminar cortical networks, can be selectively and preferentially engaged by periodic stimulation. Using leaky integrate-and-fire neuron models, we analyzed cortical circuits comprised of multiple cell-types, alongside superficial multi-layered networks expressing distinct layer-specific timescales. Our results show that mismatch in neuronal timescales within and/or between cells—and the resulting variability in excitability, temporal integration properties and frequency tuning—enables selective and directional control on synaptic connectivity by tACS. Our work provides new vistas on how to recruit neural heterogeneity to guide brain plasticity using non-invasive stimulation paradigms.

**Data Availability Statement:** The codes are available publicly on GitHub https://github.com/arefpz/neuronal_population.

## Author summary

Brain stimulation techniques, such as transcranial alternating current stimulation (tACS), are increasingly used to treat mental health disorders and to probe brain function. Despite promising results, it remains unclear how these non-invasive interventions impact both the dynamics and connectivity of neural circuits. We developed an interdisciplinary framework showing that heterogeneity in neuronal timescales, and its consequences on

**Funding:** We thank the National Research Council of Canada (NSERC GRANT RGPIN-2017-06662) (DT, AP, JL) as well as the Canadian Institute for Health Research (CIHR GRANT NO PJT-156164) (AH) for funding. AH acknowledges the support by the INRIA Action Exploratoire 'A/D Drugs'. The funders had no role in study design, data collection and analysis, decision to publish, or preparation of the manuscript.

**Competing interests:** The authors have declared that no competing interests exist.

cellular excitability and temporal integration properties of cortical neurons, may lead to selective and directional control on synaptic modifications by tACS. Differences in neuron responses resulting from timescale mismatch establishes phase- and frequency-specific tuning relationships which may be recruited by periodic stimuli to guide synaptic plasticity. We confirmed this using both intra—and inter-laminar cortical circuit models comprised of multiple cell types. Our work showcases how heterogeneity might be used to guide synaptic plasticity using non-invasive stimulation paradigms.

## Introduction

Over the past decade, there has been a growing interest in using transcranial electrical stimulation in the study of brain function and diseases [1–3]. Such findings have raised the fascinating prospect of manipulating neural activity in a controlled manner, engaging neural circuits at a functional level to manipulate cognition and treat neurobiological disorders. In addition to their reported clinical effectiveness as a treatment of major depression disorder [4, 5], epilepsy [1, 6], Parkinson's disease [7] and stroke [8, 9], transcranial electrical stimulation has given neuroscientists practical tools for examining the relationship between oscillatory neural activity and brain function through the use of rhythmic stimuli [10–13]. Periodically fluctuating electric fields, such as transcranial alternating current stimulation (tACS), are believed to enable the interference with and/or manipulation of the timing of neural signaling, impacting neural function both locally (i.e. individual cells and synapses) and globally (i.e. populations of neurons) [14–16].

Despite these advances, it remains unclear how tACS reliably modifies the connectivity—as opposed to the dynamics—of neural circuits. Indeed, beyond tACS's immediate effect on neuronal populations' activity, concurrent changes in synaptic connectivity and especially how they relate to stimulation waveforms, remains challenging to assess [17]. The entrainment of neurons' spike timing and/or phase by tACS [15, 16] suggests that temporally varying stimulation might recruit synaptic LTP/LTD, notably through spike-timing dependent plasticity (STDP)[18–21], leading to persistent changes in network dynamics and connectivity. However, tACS outcomes remain to this day notoriously ephemeral and variable: induced changes in excitability vary considerably between stimulation sites, repeated trials and subjects, oftentimes vanishing after stimulation offset [22–24] and shown to be state-dependent [25, 26]. Understanding the source of this variability must imperatively be addressed to optimize existing tACS paradigms and their effect on brain plasticity to consolidate their clinical efficacy.

One important source for such variability is the large variance in the timescales expressed by cortical neurons, resulting from diverse circuit motifs, morphology, spatial orientation and other intrinsic neuronal biophysical properties [27]. Such variations in cellular timescales not only influence how cortical neurons respond to stimulation, but further constrain the timing of neural signaling and resulting changes in synaptic connectivity. Defined as the product of neuron capacitance and membrane resistance [28], the membrane time constant (MTC) is a key biophysical parameter impacting neuronal timescale expression, varying across multiple orders of magnitude within and between cell types, cortical layers and/or brain regions [27, 29–32] as well as with input statistics [33]. The MTC influences not only shapes spike timing and phase, but also reflects the net consequence of varied biophysical attributes on cellular excitability and integration of temporally-varying stimuli [27, 34–36]. A natural consequence of this is that some neurons and/or cortical layers may be more responsive than others to rhythmic, phase- and frequency-specific entrainment. Far from limiting, this divide might

instead open up the possibility of selectively targeting cortical circuit connections using tACS with properly calibrated waveforms, capitalizing on the heterogeneity of cortical circuits to guide synaptic modifications.

With this in mind, we here reconcile the heterogeneity of cortical circuits and tACS-mediated changes in synaptic plasticity. The response of cells and populations to periodic entrainment has been well characterized, both experimentally [15, 16, 37] and with the use of computational models [10, 17, 37, 38]. We here complement and extend these by exploring how tACS at various frequencies engages brain plasticity in the presence of timescale heterogeneity. To model such heterogeneity, we randomly sampled membrane time constants from probability distributions informed by experimental data, and examined how such differences impacted the variability of neuronal responses to tACS. We deliberately set our network model in a regime where tACS elicits neuronal depolarization, to characterize how heterogeneity and tACS interact with STDP at temporal scales accessible by our simulations. We further discuss these limitations below. Our results suggest that, counter-intuitively, heterogeneous asynchronous populations might be more susceptible to efficient, directionally selective control of synaptic plasticity. Specifically, using leaky integrate-and-fire (LIF) neurons with synapses endowed with Hebbian spike-timing dependent plasticity, we explored the dependence of synaptic long-term potentiation (LTP) and depression(LTD) on tACS frequency in 1) pairs of mutually synapsed neurons; as well as 2) intra-; and 3) inter-laminar cortical circuits of excitatory and inhibitory neurons, whose MTC distributions follow layer-specific probability distributions as well as connectivity fitted on cortical physiological data [32, 39]. Exposing these different heterogeneous systems to tACS, our results show that timescale heterogeneity establishes distinct phase-relationship profiles between cells, cortical layers and stimulation waveforms. Far from limiting, such differences enable selective and directional control on intra- and inter-laminar connectivity. We hence propose tACS might capitalize on such timescale heterogeneity to guide synaptic plasticity both non-invasively and purposefully.

## Results

### Cortical timescale heterogeneity and response variability to periodic stimulation

The temporal integration and postsynaptic response of neurons to both endogenous and exogenous inputs is highly variable, notably due to different intrinsic timescales resulting from various combinations of biophysical properties, such as threshold value, rheobase, and neuronal timescales [27, 34–36]. For instance, the membrane capacitance and resistance, reflecting the net influence of various biophysical attributes, differentiate neurons in terms of the rate at which presynaptic stimuli are integrated, as well as the timescale of the output they generate. The resulting membrane time constant (MTC; $\tau_m$) varies significantly both within and across cortical layers, strongly influencing neuronal responses: neurons with smaller and larger MTC exhibit shorter and longer integration times, respectively. As depicted in Fig 1A MTCs of pyramidal neurons in cortex layers are distributed, ranging from few milliseconds in superficial layers (Fig 1A, layers I-III) to tens of milliseconds in deeper layers (Fig 1, layers V-VI). Such timescale heterogeneity results in variability in neuronal excitability and frequency tuning [40]. Endowing leaky-integrate-and-fire (LIF) neurons with such different MTCs (Eq 2; see Materials and methods and Table 1), one may see how timescale heterogeneity translates into net differences in excitability, as illustrated in Fig 1. Membrane time constants shape individual neuron response function, both without (Fig 1B1) and with (Fig 1B2) temporally varying stimulation.

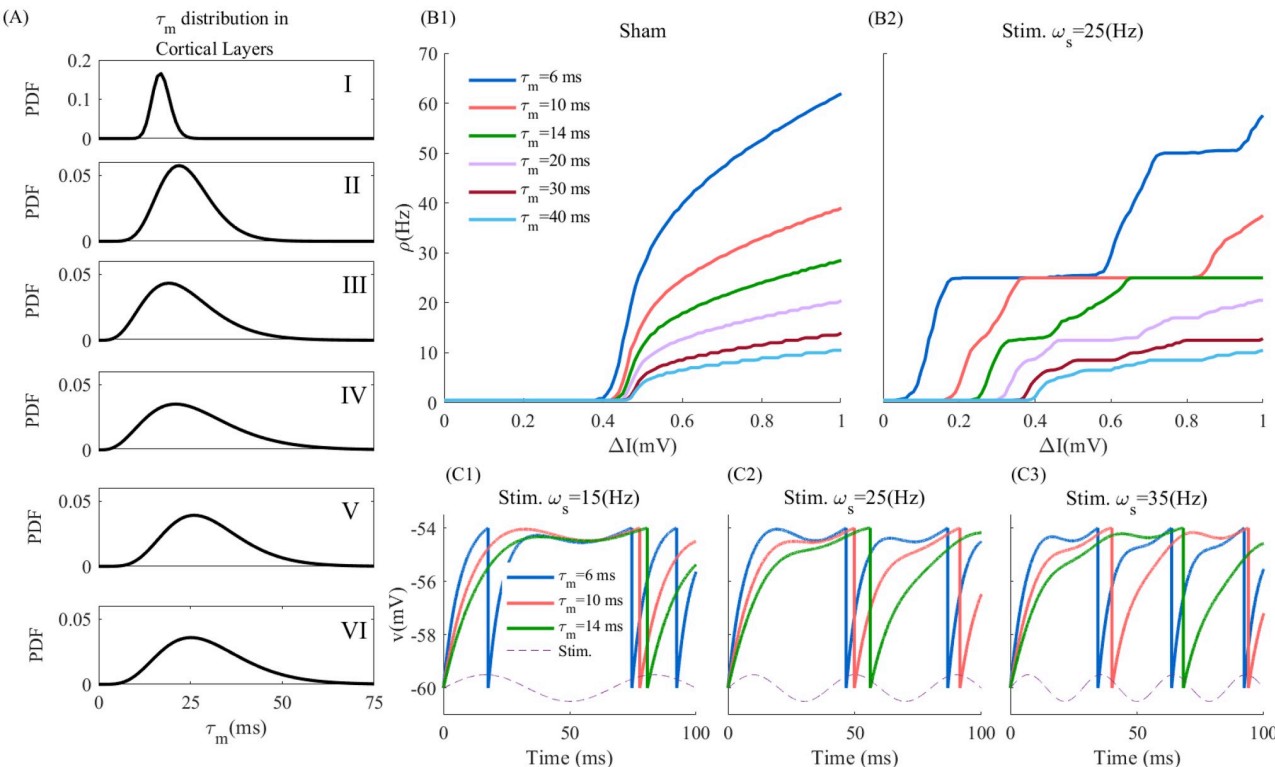

**Fig 1. Timescale heterogeneity across cortical layers.** Heterogeneity in timescale, modelled through cellular variations in membrane time constants (MTC) translates into variability in excitability. (A) The probability density functions (PDF) of membrane time constant across human cortical layers [32]. Panel (B1) and (B2) show the Frequency-Input curve in sham and the stimulation condition ($\omega_s = 25(Hz)$), respectively. The firing rate $\rho$ has been calculated from 10s simulated time series for difference input currents $\Delta I$ and averaged over 10 trials with $A_s = 0.5(mV)$, $\mu = 5.5(mV)$, $\sigma = 1(mV)$, and $\tau_{ref} = 0\ ms$ (see Eq 2). (C1), (C2) and (C3) present the temporal evolution of the membrane potential of deterministic LIF neurons at $\omega_s = 15$, 25, and 35 ($Hz$), respectively. Used parameters: $A_s = 0.5(mV)$, $\mu = 5.9(mV)$, $\sigma = 0(mV)$, and $\tau_{ref} = 0\ ms$.

The heterogeneity in timescales across cortical neurons further has important consequences from the perspective of synaptic plasticity and its potential recruitment by tACS. This can be seen in Fig 1C1–1C3, where we plotted exemplar membrane potentials and spiking responses of periodically stimulated neurons across different values of MTCs and stimulation frequencies. Faster (i.e. smaller $\tau_m$) neurons are more susceptible to entrainment across a broader

**Table 1. Neuron model parameters, synaptic inputs, and plasticity for coupled neurons.** More information is provided in the figures' caption.

| Parameters | Values | Description |
|---|---|---|
| $\tau_m$ | 4...20 ms, Varies in each figure. | Neuron membrane time constant (MTC) |
| $V_{rest}$ | -60 (mV) | Rest membrane potential |
| $g_0$ | 0.1 (mV) | Initial Synaptic weigh |
| $t_d$ | 0.5 ms | Axonal delay |
| $\tau_d$ | 3 ms | Synaptic decay time constant |
| $v_{thr}$ | -54 (mV) | Threshold value |
| $\tau_{ref}$ | 2 ms | Refractory time |
| $I_\zeta$ | $\mu = 5.5$ (mV), $\sigma = 1$ (mV) | Input current |
| $A_s$ | 0.5 (mV) | Stimulation amplitude |

range of stimulation frequencies: their spiking response remains phase locked to the stimulation even as the stimulation frequency increases. In contrast, slower (i.e. longer $\tau_m$) cells are less susceptible to entrainment as stimulation frequency increases: phase locking at lower stimulation frequencies vanishes as the stimulation frequency increases.

Membrane time constant heterogeneity, either across cells or cortical layers, shapes phase relationships between neuronal responses and tACS stimulation, resulting in various degrees of susceptibility to entrainment. This is illustrated in Fig 2, where we plotted MTC-induced phase relationships between neuronal spiking of LIF neurons with varying MTCs under periodic stimulation at various frequencies. While increasing either MTCs and/or stimulation frequency suppresses the net neuron's firing response, they also impact the slope of the spiking phase (Fig 2B1–2B3). Ought to the manifest variations of timescales observed within and across cortical layers (MTCs; c.f. Fig 1), the results of Fig 2 suggest that stimulation-induced changes in synaptic plasticity might be both neuron-, layer- and stimulation-frequency dependent. We note that this type of behavior is not exclusive to the LIF model neurons portrayed here, being observed in more complex neuron models as well (see [41]).

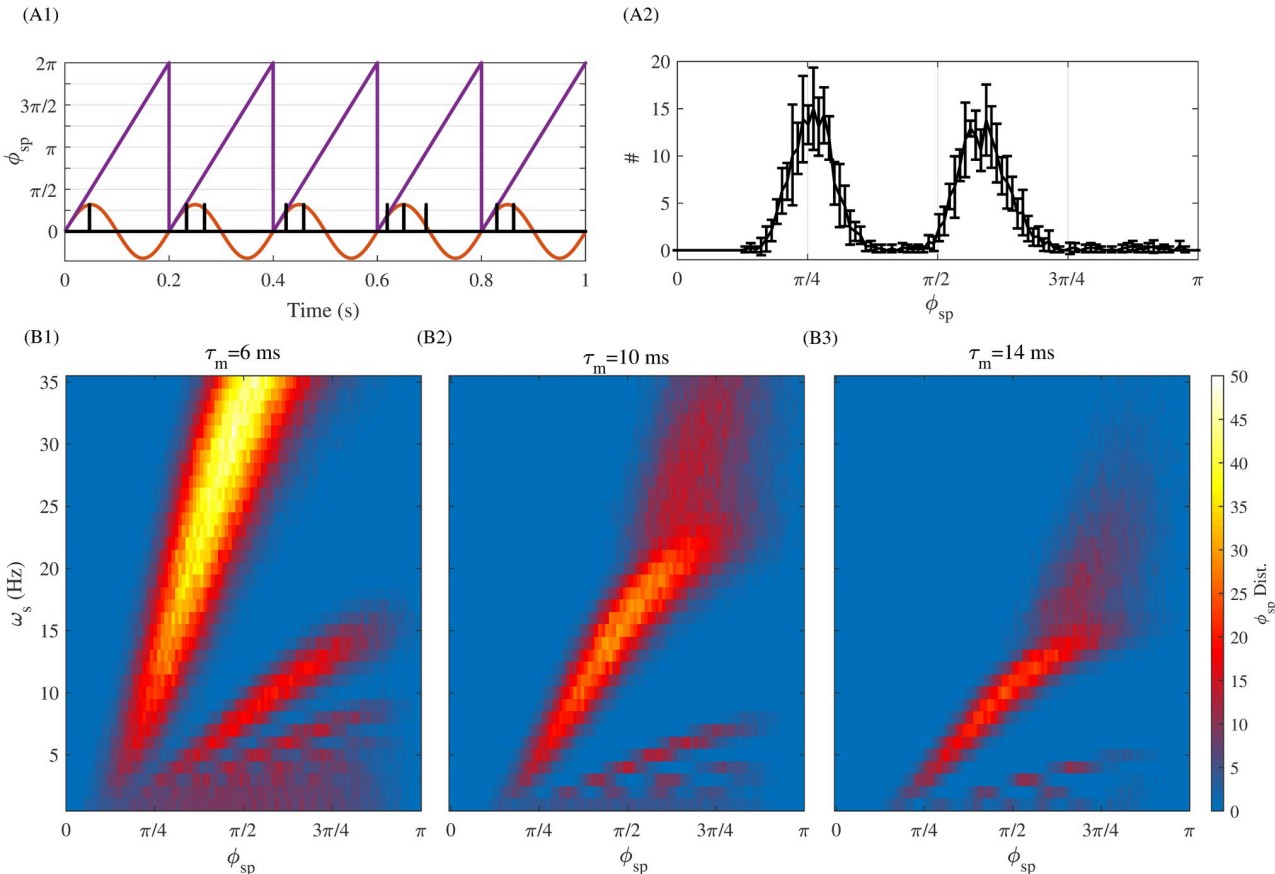

**Fig 2. Neuronal timescales shape phase-relationships between neuron responses and periodic stimuli.** (A1) Exemplar spiking and phase of an individual neuron with MTC ($\tau_m$) of 10 ms under periodic stimulation. The red line is the periodic stimulation with $\omega_s = 5(Hz)$ and the purple line shows the phase of stimulation. The black lines locate neuronal spiking times computed by integrating the single neuron LIF-model. (A2) The resulting phase histogram of spiking activity of the LIF neuron shown in (A1). The vertical axis gives the number of spikes that occurred at appropriate stimulation phase as of x-axis. Note that this is bimodal because the stimulation (at $\omega_s = 5Hz$) depolarising the neuron enough to make it fire repeatedly during one stimulation oscillation period. (B1)-(B3): Phase response relationships to periodic stimulation of increasing frequency (for different MTC $\tau_m$ = 6, 10, 14 ms). These results were obtained by taking the average over 10 independent trials, over 10s of simulation time. Here we used $\mu = 5.5(mV)$, $\sigma = 1(mV)$ and $A_s = 1(mV)$.

## Neuronal timescale mismatch guides plasticity in individual synapses

We first examined the simplest network motif expressing timescale heterogeneity: we considered a pair of mutually synapsing LIF excitatory neurons, while investigating how MTC mismatch shapes the magnitude and directionality of tACS-induced synaptic modifications at the level of an individual synapses. We hypothesized that MTC disparity in such a two neuron network, and the resulting timescale separation, can be recruited using tACS to selectively and directionally potentiate and/or depress synapses. To model plasticity, we endowed each synapse in this simple model with Hebbian spike-timing dependent plasticity (STDP) [40, 42, 43] (see Materials and methods). As synaptic weight modifications follow spike timing differences (i.e. $\Delta T$) between pre- and post-synaptic neurons, membrane time constant diversity and its consequence on spike phase will influence synaptic modifications induced by periodic stimulation. To this end, stimulation amplitude was deliberately increased to induce depolarization across a wide range of MTCs to expose the interaction between tACS and spiking phase and their effect on synaptic plasticity. We first set the MTC value of the pre-synaptic neuron to $\tau_m^{(1)} = 10$ $ms$, while the MTC of the post-synaptic neuron was set to either $\tau_m^{(2)} = 6$ $ms$, 10 $ms$ or 14 $ms$, respectively. Distributions of spike timing differences $f(\Delta T)$—between any pair of pre- and post-synaptic spikes, $\Delta T = t_{sp}^{post} - t_{sp}^{pre}$ are plotted in Fig 3A1 and 3A2. Set in the asynchronous regime, both neurons in the network were stimulated identically. We note that throughout, the Hebbian STDP mechanism and its associated parameters (see Materials and methods) remained unchanged.

As shown in Fig 3A1, in the absence of stimulation (i.e. sham), the spike timing difference distribution is flat: the asynchronous firing of both pre- and post-synaptic neurons prevents any preferential and/or directional synaptic modification. This indicates that synaptic weights between neurons remain on average, constant. However, by applying tACS, MTC mismatch between the neurons, and the resulting disparity in their spiking phase responses, polarizes the spike timing difference distribution (Fig 3A2 when $\omega_s = 25(Hz)$), eliciting preferential directions in synaptic modification. As can be seen in Fig 3A3, a net gain and directionality of stimulation-induced changes in synaptic connectivity was found to be dependent on 1) the mutual arrangement of pre- versus post-synaptic neurons' MTCs; and 2) on the specific choice of stimulation frequency. For a pre-synaptic neuron with $\tau_m = 10$ $ms$, stimulation-induced synaptic potentiation could be observed when coupled to a slower (i.e. $\tau_m = 14$ $ms$) post-synaptic cell. The opposite happens whenever coupled with a faster (i.e. $\tau_m = 6$ $ms$) post-synaptic neuron, and synaptic depression can be observed. The magnitude of synaptic potentiation and/or depression was further found to scale with tACS stimulation frequency. To see this, we fixed the pre-synaptic MTC to $\tau_m^{(1)} = 10$ $ms$ while systematically varying $\tau_m^{(2)}$ over values ranging from $4ms$ to 20 $ms$ and across stimulation frequencies up to $50(Hz)$, examining resulting synaptic modification amplitude and directionality. As can be seen in Fig 3B, the difference between the neurons' MTCs not only supports the previous results but further shows that net changes in synaptic weights (i.e. potentiation or depression) and their direction can be tuned in a stimulation-frequency specific manner. Taken together, the above analysis indicates that MTCs disparity in a two-neuron network motif, through resulting change in spike timing difference distribution, enables selective, direction and stimulation frequency-specific changes in synaptic connectivity.

## Directional tACS-induced synaptic plasticity in a heterogeneous cortical layer

To generalize our observations, we analyzed a sparse network of $N = 10,000$ LIF neurons, modeling the response of a neuronal population, representing a single cortical layer, to tACS

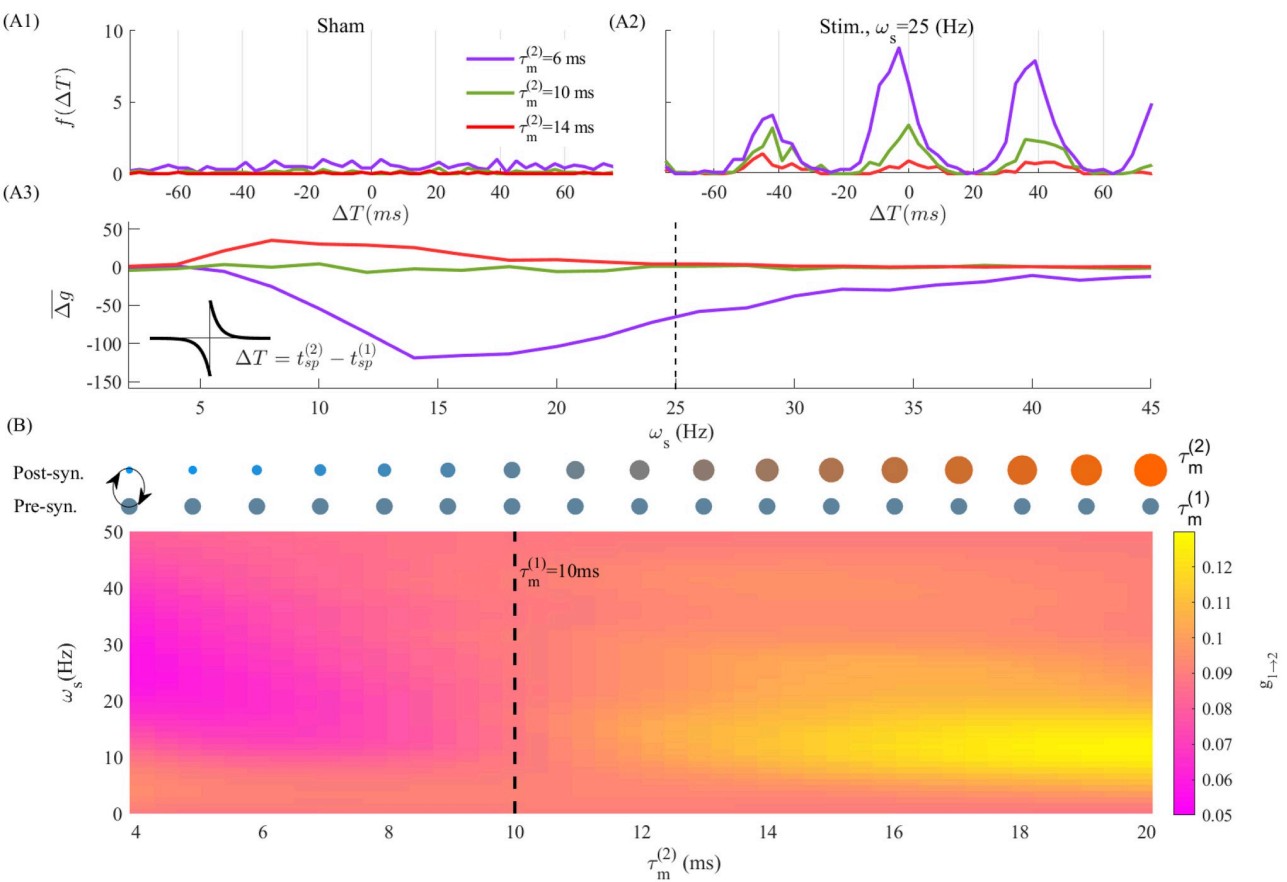

**Fig 3. Timescale mismatch between two synapsing neurons yields direction- and stimulation-frequency dependent synaptic plasticity.** In (A1) and (A2), we plotted the distribution of spike timing difference, $f(\Delta T)$, between neuron (2) and neuron (1), where $\Delta T = t_{sp}^{(2)} - t_{sp}^{(1)}$, at two stimulation frequencies $\omega_s = 0, 25 (Hz)$, respectively. This distribution was obtained over $10s$ simulation time. Here $\tau_m^{(1)} = 10ms$ and the MTC of neuron (2) is changing from $\tau_m^{(2)} = 6ms$, $10ms$ to $14ms$. For instance, the purple line indicates the distribution of $\Delta T$ between post-synaptic neuron (2) and pre-synaptic neuron (1). In (A3), we plotted the mean synaptic weight change $\overline{\Delta g}$, cf. Eq (7) in the *Materials and methods* section. The dashed line denotes the stimulation frequency used in (A2) to calculate the $\Delta T$. We schematically relate the MTC of neurons to the size of circles in (B) top panel, i.e. a larger diameter represents a larger MTC. The top and bottom circles denote the MTC of neuron (2) and neuron 1, respectively. The heatmap plot in (B) bottom, shows the synaptic weight modifications between coupled excitatory neurons (i.e. synapse from neuron (1) to neuron (2), $g_{1 \rightarrow 2}$), over different stimulation frequencies (y-axis) and second neuron's MTC (x-axis). The MTC of neuron (1) kept at $\tau_m^{(1)} = 10ms$ (indicated by the vertical dashed line). The initial value of synaptic weight between neurons is $g_0 = 0.1 (mV)$. (see Table 1).

(see Materials and methods). We characterized the influence of timescale heterogeneity amongst cell types on intra-laminar synaptic recruitment by periodic stimulation and the dependence on its frequency. Such heterogeneity was introduced in the network by randomly sampling individual MTC from independent normal, positive definite distributions with mean $\mu_{\tau_m} = 10\ ms$ and standard deviation $\sigma_{\tau_m} = 3\ ms$ (see Fig 4A). This approach captures intra-laminar variability both between (i.e. excitatory and inhibitory) and within (i.e. subtypes of pyramidal and interneurons) cell types, collectively expressing heterogeneous MTC and resulting integration properties.

tACS efficacy has been shown to be highly state-dependent, enhanced for regimes of asynchronous neural activity where endogenous oscillations are suppressed [25, 26]. We hence focused on a regime of asynchronous spiking to quantify tACS-induced synaptic changes in the absence of endogenous oscillations. To do this, we set the network's parameters (see Table 2) so that the network resides in a balanced state, for which neurons exhibit

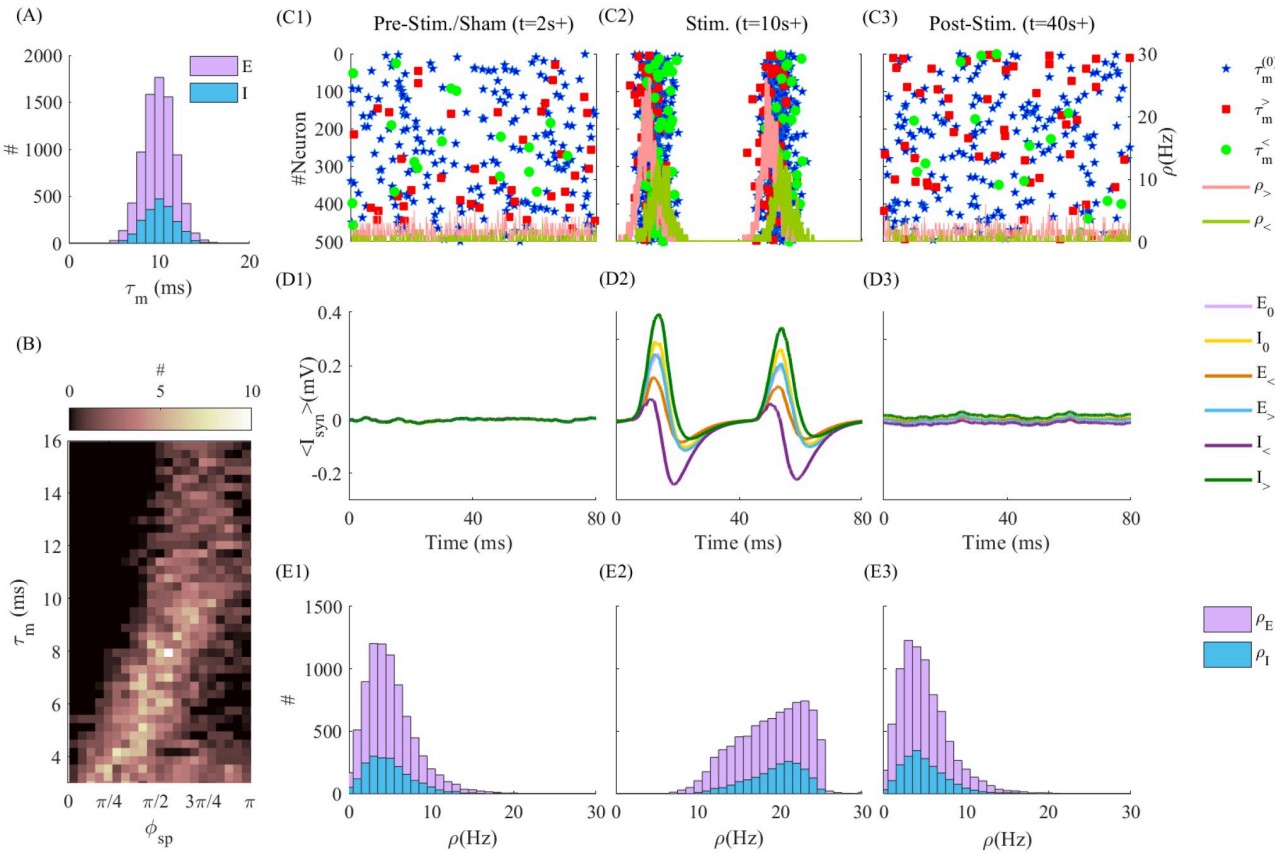

**Fig 4. tACS modulation of a cortical layer network of heterogeneous excitatory and inhibitory neurons in the asynchronous state.** (A) The MTC distribution for excitatory and inhibitory neurons with $\mu_{\tau_m} = 10$ $ms$ and $\sigma_{\tau_m} = 3$ $ms$. (B) shows the number of neurons with MTC as y-axis locked to phase of stimulation (with $\omega_s = 25(Hz)$) as x-axis for one second of simulation time. (C1) to (C3) present the time evolution of spiking activity of 500 randomly selected neurons (250 E and 250 I neurons) for Pre-stim./Sham, Stim. and Post-stim. periods, respectively. The blue stars represent the spikes of neurons with $\tau_m^{(0)} \in [9.5\ 10.5]ms$, the red squares and line (green circles and line) correspond to the spikes and firing rate of neurons with $\tau_m^< \in \{\tau_m \leqslant 8ms\}$ ($\tau_m^> \in \{\tau_m \geqslant 12\}$), respectively. (D1) to (D3) show the average of synaptic input across different neurons categories shown in legend, in Pre-stim./Sham, Stim. and Post-Stim. periods, respectively. i.e. $E_<$ is the mean synaptic inputs averaged over excitatory neurons with $\tau_m^<$. (E1) to (E3) show the firing rates of excitatory and inhibitory neurons in Pre-Stim./Sham, Stim. and Post-Stim. periods, respectively.

asynchronous irregular (AI) activity (see Fig 4C1, 4C2 and 4C3) [44]. Stimulation amplitude was set at $A_s = 1mV$ to ensure that neurons are strongly entrained to the stimulation. This amplitude is further within the range of the values observed experimentally [21, 45]. All cells were stimulated identically. We note that throughout, STDP (i.e. Eq 6, see Materials and methods) was effective both within the excitatory population (i.e. $E \rightarrow E$) as well as between excitatory and inhibitory population (i.e. $E \rightarrow I$ and $I \rightarrow E$).

To quantify directionality and associated changes in synaptic weights, we grouped neurons according to their timescales and compared how tACS modified their mutual synaptic connectivity. Specifically, we compared the (changing) connectivity within and between groups with different timescale (i.e. MTC) statistics i.e., fast responding ($\tau_m^{E,I<} \in \{\tau_m \leqslant 8\ ms\}$) versus slowly responding ($\tau_m^{E,I>} \in \{\tau_m \geqslant 12\ ms\}$) neurons, as well as with cells whose MTC resides close to the chosen mean value($\tau_m^{E,I(0)} = 10 \pm 0.5\ ms$). Here (E) stands for excitatory and (I) for inhibitory neurons.

We examined the response of this cortical layer network before, during, and after stimulation while quantifying associated synaptic modifications. Specifically, the network's dynamic

**Table 2. Parameters of the populations.** In Fig 6 the number of neurons are N = 12000 which is distributed among 4 equal cortical layers. For more information about populations, see appropriate caption and Materials and methods section.

| Parameters | Values | Description |
|---|---|---|
| $N_E$ | 8000 | Number of excitatory (E) neurons |
| $N_I$ | 2000 | Number of inhibitory (I) neurons |
| $P_{xy}$ | 10%, $x, y \in [E, I]$ | Connectivity probability among neurons |
| $\tau_m$ | $\mu_{\tau_m} = 10, \sigma_{\tau_m} = 3\ ms$ | Neuron membrane time constant (MTC) |
| $V_{rest}$ | -60 ± 0.2 (mV) | Resting membrane potential |
| $g_0^{E \to E}$ | 5E-5 (a.u.), $\sigma_g = 0.1g_0$ | Initial Synaptic weigh among E to E neurons |
| $g_0^{E \to I}$ | 5E-5 (a.u.), $\sigma_g = 0.1g_0$ | Initial Synaptic weigh among E to I neurons |
| $g_0^{I \to E}$ | 25E-5 (a.u.), $\sigma_g = 0.1g_0$ | Initial Synaptic weigh among I to E neurons |
| $g_0^{I \to I}$ | 25E-5 (a.u.), $\sigma_g = 0.1g_0$ | Initial Synaptic weigh among I to I neurons |
| $g_{min}$ | $0.01 \times g_0$ | Minimum Value of synaptic weight |
| $g_{max}$ | $2 \times g_0$ | Maximum value of synaptic weight |
| $E_{syn}$ | E = 0 mV, I = -85 mV | Reversal potential |
| $t_d$ | 0.5–1 ms | Axonal delay |
| $\tau_r$ | 0.5 ms (AMPA), 0.5 ms (GABA$_a$) | Synaptic rise time constant |
| $\tau_d$ | 3 ms (AMPA), 5 ms (GABA$_a$) | Synaptic decay time constant |
| $v_{thr}$ | -54 (mV) | Threshold value |
| $\tau_{ref}$ | 2 ms | Refractory time |
| $I_\zeta$ | $\mu = 5.5\ (mV)$ and $\sigma = 1\ (mV)$ | Mean input current and noise SD. |
| $A_s$ | 1 (mV) | Stimulation amplitude |

was simulated for a period of $t = 60$ seconds (simulation time) and subjected to a tACS-like periodic stimulation during an epoch ranging from $t = 4s$ to $t = 40s$. Representative responses of neurons amongst the groups defined above, before ($t = 2s$), during ($t = 10s$) and after ($t = 40s$) tACS stimulation at $\omega_s = 25(Hz)$ (we generalize our conclusions to a broader range of stimulation frequencies later) are plotted in Fig 4C1, 4C2 and 4C3 respectively. In absence of stimulation (Fig 4C1; sham), asynchronous firing can be observed across all neurons, irrespective of MTC differences. Set in the balanced state, the net synaptic input taken across all cells in the network is close to zero (Fig 4D1, resulting in weak and uncorrelated firing rates (Fig 4E1). The average firing rates in sham period were accordingly distributed around a mean of $5(Hz)$, for both excitatory and inhibitory neurons. In contrast, in presence of stimulation, strong amplitude, phasic responses can be observed following tACS entrainment. Due to MTC heterogeneity (Fig 4B), firing phase was expectedly found to vary across the network in an MTC-specific way in presence of stimulation. Similarly to what was observed in Fig 2, phases at which individual cells are responding increases with MTC, confirming that stimulation engaged different neurons differently in networks as well. Such responses where further found to be timescale group-specific: Indeed, as can be seen in Fig 4C2, the phase of firing and amplitude were found to vary with MTCs. This can be seen from the mean synaptic inputs, whose phase and amplitude varied with respect to cell type (E vs I) as well as MTC characteristics. Firing rates of both excitatory and inhibitory populations also increased compared to sham (Fig 4E2). We highlight that such increase in firing rate—which is not consistently observed experimentally [15, 16, 46]—is here a direct consequence of our choice of tACS amplitude, causing both depolarization and entrainment. Similar results, obtained with smaller tACS amplitudes, were not accompanied by significant changes in firing rates (see S1 Fig).

While network activity reverts to pre-stimulus state after stimulation (see Fig 4C3, 4D3 and 4E3), the mean synaptic input do differ compared to the pre-stimulation/sham period, indicative of changes in intra-laminar synaptic connectivity between groups –and neurons—with differing MTCs statistics.

Our results indicate that timescale mismatch plays an important role in the selective recruitment of synaptic plasticity by tACS. Indeed, resulting effects of MTC heterogeneity on intra-laminar synaptic modifications due to tACS entrainment are shown in Fig 5. In Fig 5A1, 5A2 and 5A3 we plotted the distribution of synaptic weight between and amongst excitatory and inhibitory neurons with near average MTCs (i.e. 10 $ms$; $\tau_m^{E,I^{(0)}}$), synapsing onto neurons with either faster ($\tau_m^{E,I^<}$) and/or slower ($\tau_m^{E,I^>}$) MTCs. Due to MTC mismatch, synaptic weight distributions are shifted in a MTC- and stimulation frequency-specific dependent manner. Synapses between slower (i.e. longer MTC) and faster (i.e. shorter MTC) responding neurons were potentiated, and were otherwise depressed, irrespective of cell type. This symmetry between cell type results from identical MTC distributions for both excitatory and inhibitory populations, although they encompass variations within each cell-type. In the

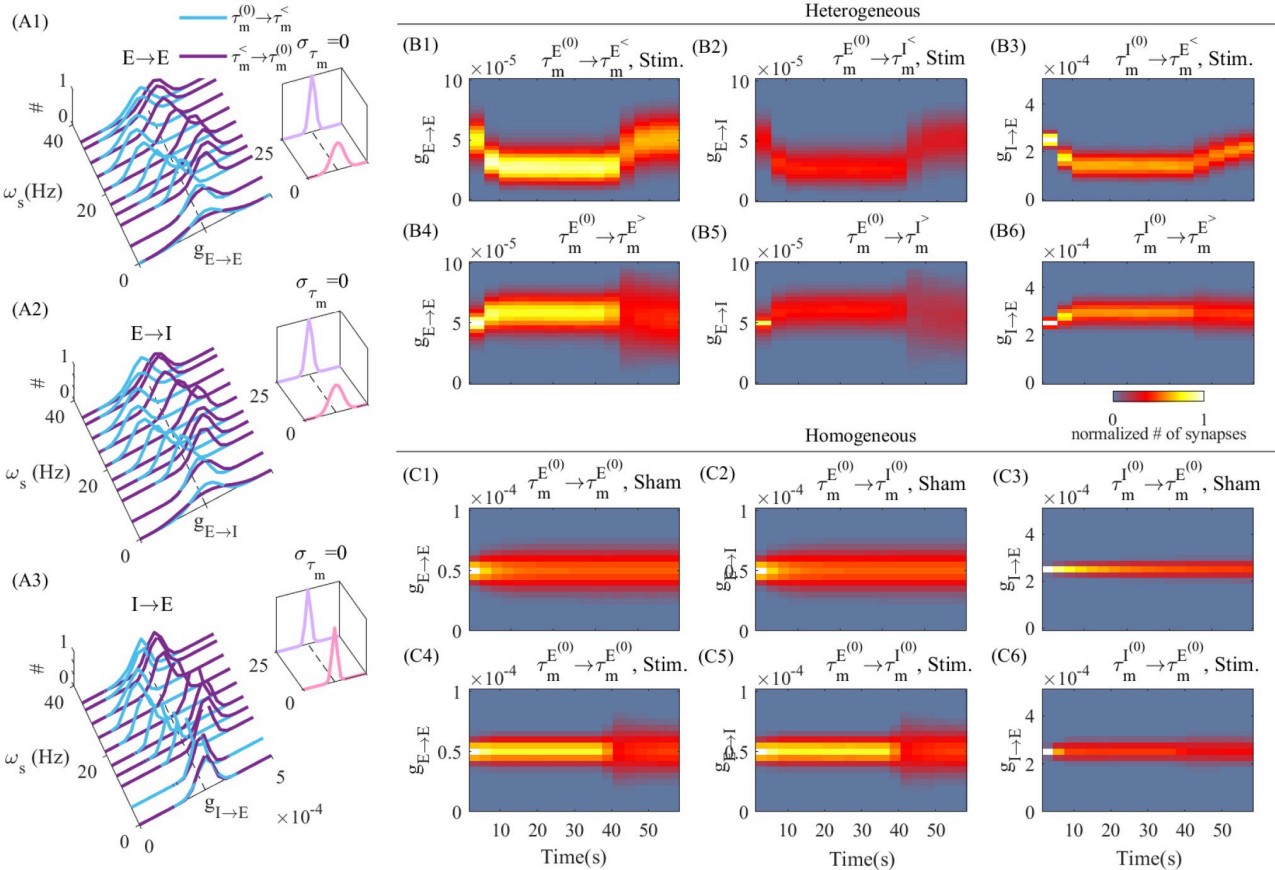

**Fig 5. tACS-induced changes in intra-laminar synaptic connectivity.** (A1)-(A3) Distribution of synaptic weights among excitatory and inhibitory neurons. In each inset plot we plotted the homogeneous case when all neurons have the same MTC, $\sigma_{\tau_m} = 0$ at two stimulation frequencies, $\omega_s = 0, 25$ ($Hz$). In (B1)-(B6) and (C1)-(C6) we plotted the change of synaptic weight distributions when the MTCs were selected heterogeneously and homogeneously, respectively. (B1,B2,B3) and (B4,B5,B6) show the time evolution of distribution of synaptic weights for synapses from neurons with $\tau_0 = 10$ $ms$ to neurons with $\tau_m^<$ and $\tau_m^>$, respectively when $\omega_s = 25(Hz)$, $A_s = 1(mV)$. The stimulation is turned ON at $Time = 4(s)$ and turned OFF at $Time = 40(s)$. On top of each plot the category of synaptic weights is mentioned, i.e. (B1) shows the synaptic weights among excitatory neurons with $\tau_m^{(0)}$ to neurons with $\tau_m^<$. In (C1,C2,C3) and (C4,C5,C6), we plotted the synaptic weights changes in Sham and Stim. states (inset plots in A1-A3) respectively. The colorbar encodes the relative frequency of synapses with respect to synaptic weights as vertical axis.

homogeneous case, in which all neurons possess the same MTCs ($\sigma_{\tau_m} = 0$), synaptic weights remained stable, and tACS entrainment does not lead to any preferential changes in weight distribution. These results mirror those observed in the simple two-neuron motif examined in Fig 3. To quantify the evolution of synaptic weight distribution before, during and after tACS entrainment, we plotted in Fig 5B1–5B6 the time-averaged synaptic weights for the illustrative case where the stimulation frequency is set at $\omega_s = 25(Hz)$. This illustrative frequency was selected as it yield the maximal change in synaptic weight distribution (Fig 5A1, 5A2 and 5A3). After stimulation onset ($t = 4s$), entrainment results in directional synaptic weight modification: synapses between average and fast neurons ($\tau_m^{E,I(0)} \to \tau_m^{E,I^<}$) are depressed, while those linking average and slow neurons ($\tau_m^{E,I(0)} \to \tau_m^{E,I^>}$) are potentiated, irrespective of cell type. Such changes saturate due to a net cancellation between potentiation and depression. At stimulation offset ($t = 40s$), synaptic weights slowly converge back to their pre-stimulation values, although displaying increased variance compared to baseline. Whenever MTC heterogeneity is removed, ($\sigma_{\tau_m} = 0$), no synaptic modification could be observed (Fig 5C1–5C6), either without (Fig 5C1, 5C2 and 5C3) or with tACS stimulation (Fig 5C4, 5C5 and 5C6). In both of these cases, the distribution of synaptic weights remained around the baseline value. Taken together, these findings indicate that timescale variations through MTC heterogeneity enables tACS to guide intra-laminar synaptic plasticity in a directional and frequency-specific manner.

## tACS engagement of inter-laminar connectivity

Motivated by these results, we questioned whether timescale heterogeneity, and the resulting variability in neurons' response to periodic stimuli, could be solicited to engage and modify inter-laminar connectivity in superficial cortical networks, which are preferentially recruited during tACS [37, 47–49]. Cortical layers are populated by both pyramidal cells and interneurons with diversified biophysical profiles [32, 39]. To examine this possibility, we extended our analysis using a model of recurrently connected neuronal population. Individual cortical layers, and their associated intra-laminar connectivity, were modeled as recurrently connected populations of excitatory and inhibitory LIF neurons, using the same parameters and sitting in the same dynamical regime as reported above. We connected this multi-layer cortical model using known inter-laminar projections between cortical pyramidal cells in the primate brain [39, 50] (Fig 6A. See Materials and methods for model description.). Layer-specific timescales, reflected through distinct MTC distributions, were imported from the Allen institute cell database [32](i.e. Fig 1A). We assumed that the modulatory effect of tACS on the neurons' membrane potential remained the same across layers.

Consistent with our previous findings, stimulation enabled the selective potentiation and/or depression of inter-laminar connectivity in a tACS frequency- and MTC-dependent manner. In Fig 6B, we plotted the net changes in synaptic weight distributions amongst excitatory cells whose synapses traverse different layers, and across different tACS frequencies. Here the synapses with the minimum value of $5ms$ difference in the coupled neurons' MTC have been analyzed. Inter-laminar connectivity from layers expressing statistically shorter MTC (e.g. layers III-IV) towards those with longer MTC (i.e. layers V-VI) were potentiated, while the reverse directions were depressed. Maximal effects could be seen at stimulation frequencies ranging from the theta to the beta range (i.e. $5 - 15Hz$), and were weak outside that range. The net difference between the sum of synaptic weights, taken before and at the end of the stimulation epochs, also reveals directional, layer-specific synaptic modifications (Fig 6C). Both the magnitude and the direction of overall synaptic weights depend on the stimulation frequency. These imply changes in inter-laminar connectivity: For instance note the sign and strength of

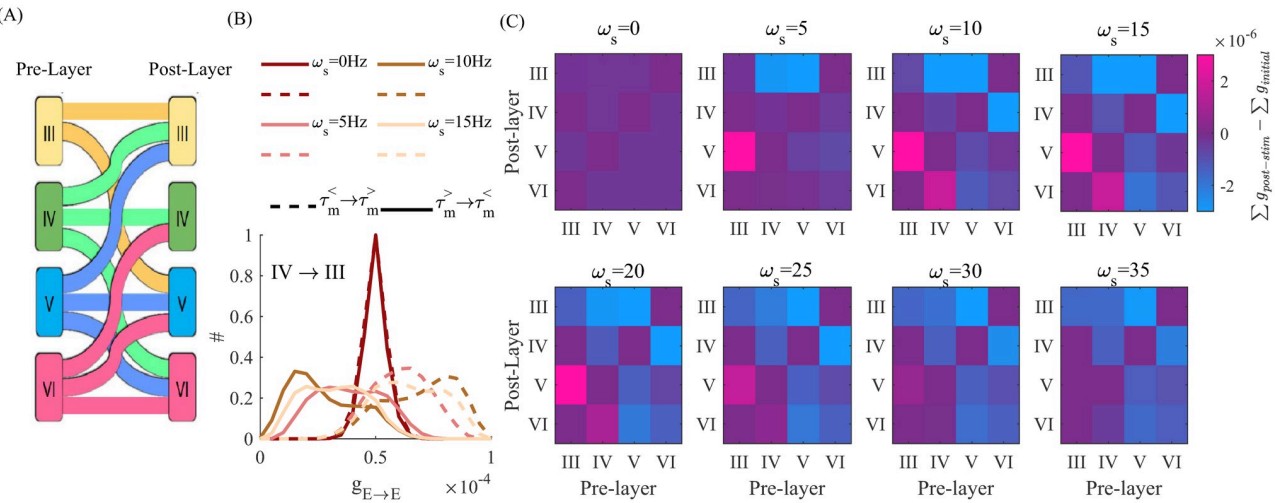

**Fig 6. Modifications of cortical layers connectivity due to tACS.** (A) Illustrative schematics of inter-laminar excitatory connections. (B) Synaptic modifications among excitatory neurons between cortical layers IV to III. The solid (dashed) lines show the histogram of synaptic weights among neurons with $\tau_m^>$ ($\tau_m^<$) to neurons with $\tau_m^<$ ($\tau_m^>$). The minimum MTC difference across neurons is 5ms. The colors indicate the frequency of stimulation, $\omega_s$. (C) Corresponds to the net difference in inter- and intra-laminar synaptic weights induced by tACS at different stimulation frequencies. The heat map encodes for potentiation (positive values) or depression (negative values). For details of the network connectivity and its parameters see the Materials and methods section.

overall synaptic weight among connections from IV to VI: while the stimulation at 15Hz potentiate the connections, stimulation at 35*Hz* depresses the connections.

## Discussion

Membrane time constants (MTC) vary both within and across cell types, influencing cellular excitability [27, 29–32] as well as the neuron's agility with respect to incoming stimuli [51]. Our simulations provide a proof-of-principle that timescale variations observed between cells, within and across cortical layers, allow tACS to guide intra- and inter-laminar connectivity, and that resulting modifications in synaptic connectivity are frequency-specific. Ought to this diversity of timescales, the distinct responsiveness of neurons to tACS gives us the possibility of engineering neural connectivity, either locally or among brain regions. Indeed, for a given tACS frequency, neurons with smaller MTC are potentially capable of responding at higher firing rates, becoming effective dynamical hubs [52] and promoting the potentiation of synapses they project to. Such synaptic changes might hence promote the formation of new stimulation-induced routes for information transfer. While our model does not reproduce nor predict the persistence of those synaptic modifications beyond stimulation offset due to the simplicity of our plasticity rule, we however provide insight regarding how to reliably engage cellular and inter-laminar connectivity. Transcranial electrical stimulation paradigms such as tACS preferentially target superficial (multi-layered) cortical networks. Our results hence provide strong support for the pursuit of non-invasive stimulation techniques not only to modulate neural activity, but to engage and guide synaptic plasticity. More detailed modeling approaches are warranted to better characterize the complex interplay between individual neurons' biophysical features, mutual synaptic connectivity, and the potential influence of brain stimulation.

Seminal experimental [15, 16] and computational [46] studies have reported no significant increase in firing rate during tACS, in contrast to our simulations where elevated spiking was observed. Experimental results remain however conflicted [21, 53–55] and the nature of such

disparity remains to be fully elucidated. We have deliberately conducted our simulations in a regime of high tACS amplitude sufficient to cause depolarization of neurons, to amplify resulting STDP-induced synaptic changes and evaluate their dependence on MTC disparity at temporal scales accessible by our simulations. In this regime, increased stimulation amplitude engage and modulate the neurons membrane potential and firing rate across a wider rang of MTCs [46, 53], where changes in firing rates are known to engage synaptic plasticity [21, 56]. We have nonetheless extended our analysis to low tACS amplitude regimes to evaluate the robustness of our findings (see S1 and S2 Figs). These additional simulations show that stimulation-induced changes in synaptic weights scale with tACS amplitude. Despite no significant change in firing rate, spike timing phase-locking lead to small MTC-specific synaptic modifications (cf. S1 and S2 Figs). These observations are in line with results previously reported [15, 16, 46] and suggest that the effective tACS amplitude (measured at the level of individual neurons) represents an important source of variability [57]. By construction, our model is devoid of many anatomical and experimental constraints that are known to hinder tACS efficacy and overall signal-to-noise ratio on membrane potential, such as skull shunting [57, 58] and cellular orientation [16, 59, 60]—which certainly influences the (effective) magnitude of neural responses to tACS. Importantly, although it is certainly possible to entrain neural population at small tACS amplitudes without altering their net firing rate, our simulations suggest that such approaches may fail to modify synaptic weights through STDP (see S1 Fig). We further note that in regimes of asynchronous irregular activity (like the one considered in our work), stimulation may not recruit resonance [10], in contrast to what would be expected in oscillating network where low amplitude stimulation [37, 53] may reliably modulate network activity and alter firing rates [10, 61]. Despite these limitations, our study reveals selective and directional synaptic modifications that are both heterogeneity- and stimulation frequency specific. We believe those results represent an important conceptual step forward the optimization of tACS (which historically suffers from severe inter-trial and inter-subject variability, and whose clinical efficacy/utility has been debated [57]) as well as other neuromodulatory paradigms (such as TMS or intracranial stimulation, in which firing rates do vary significantly during stimulation [62–64]). We believe these conclusions are important—and further illustrate the relevance of computational modelling in exploring parameter ranges that are not (currently) accessible in experiments, notably to guide new avenue of investigation.

Our results have been obtained for neurons and/or networks residing in the asynchronous irregular state (AI) [44], for which endogenous oscillations are suppressed or absent. While the interplay between tACS periodic waveforms and endogenous oscillations has been the primary focus of most studies [10, 15–17, 37], its efficacy has been shown to exhibit a strong state-dependence, maximized for regimes of irregular, asynchronous neural activity [25, 26], not oscillatory ones. Given that correlated neural activity reflects redundant, information-poor states [65, 66]: the recruitment of, interference with, and/or amplification of endogenous neural oscillations might hence prevent effective control on brain plasticity and instead promote tACS outcome variability. Our results indeed suggest that intrinsic rhythmic activity, such as those that would arise through recurrent interactions, limits the ability of exogenous stimulation (i.e. tACS) to engage synaptic connectivity. Indeed, the global phase alignment resulting from endogenous oscillatory activity leads to a suppression of time-scale differences amongst neurons. We hypothesize that this form of competition between endogenous and exogenous entrainment could be at play in the state-dependent effects of tACS reported both experimentally [25] as well as computationally [26]. However, tACS does emulate a state of synchronization which has been proposed to play an important role for memory formation and consolidation [67]. Such oscillatory modulation, and its interference with endogenous rhythms, may explain part of its effectiveness in the treatment of a wide range of neurological

and neuropsychiatric disorders. As such, the diversity and organization of MTCs, as well as a other cellular biophysical parameters, is ought to play a fundamental functional role in the modulation of specific intra and extra-laminar connectivity, unfolding yet another dimension by which transcranial stimulation might be used in the clinics. Such considerations are left for future work.

While insightful, our model nonetheless suffers from limitations. First, we considered neuronal populations with random intra-laminar connectivity [68], which remains a simplification for the cortical connectivity observed experimentally [69]. Moreover, it is important to note that axonal and/or synaptic delays may influence the stimulation-phase relationships and hence impact STDP-induced synaptic modification [70] while retaining the obtained results qualitatively. In addition, our study considers the summed MTC distribution to account for sub-cell-type differences. Further work should introduce cell-specific variations in MTC to examine the potential influence of tACS on recruiting excitatory-inhibitory connectivity. We should also mention that our results are valid in the context where stimulation-induced perturbations are salient enough relative to recurrent pre-synaptic current i.e. entrainment of membrane potential is required. This was guaranteed in our model by selecting appropriate tACS amplitude while the network expresses—due to the choice of parameters—asynchronous irregular activity. Lastly, our results are limited to the weak connectivity regime, where plasticity follows a Hebbian STDP rule. Moving forward, the inclusion of a larger variety of synaptic plasticity mechanisms (notably between cell types) [42] might further enhance the physiological relevance of our model and further extend the dynamic range expressed by the network, both with and without tACS.

## Materials and methods

### Spiking neuron model

Our simulations are based on standard Leaky-Integrate-and-Fire (LIF) excitatory and inhibitory neurons [28]. This model reliably simulate the response of neurons to incoming input current based on neuron's MTC. The following differential equation describes the evolution of the neuron's membrane potential as a function of incoming input

$$\tau_m \frac{dv}{dt} = (V_{rest} - v) + I_\zeta + I_{syn} + I_s, \tag{1}$$

for which the solution can be written as

$$v(t) = V_{rest} + v_0 e^{-t/\tau_m} + \frac{1}{\tau_m} \int_{-\infty}^{t} e^{-(t-t')/\tau_m} (I_\zeta(t') + I_{syn}(t') + I_s(t')) dt', \tag{2}$$

where $\tau_m$ is the MTC, $v$ is the membrane potential with initial value $v_0$, $V_{rest}$ is the resting membrane potential, $I_\zeta$ is white noise input current with mean value $\mu$ and standard deviation $\sigma$. The variable $I_{syn}$ represents the synaptic input. The $I_s$ represent the tACS-induced current, which we assumed has a sinusoidal form, i.e. $I_s = A_s \sin(2\pi\omega_s t + \theta)$, where $A_s$ is the amplitude of the periodic signal, $\omega_s$ is the angular frequency, and $\theta$ is the phase of stimulation. We solved Eq 1 with Euler–Maruyama method and time step $dt = 0.1 ms$. When the membrane potential crosses the threshold value, $v_{thr} = -54 (mV)$, a spike is elicited, and the membrane potential resets to its resting value $V_{rest} = -60 \pm 0.2 (mV)$ for a period of $\tau_{ref} = 2\ ms$ representing the neuronal refractory period Note that larger refractory periods alter the firing rate distribution, but the results remain consistent. We used the given parameter values in Table 2, which are in the physiological range [40, 71]. Without loss of generality, it is possible to use re-scaled and dimensionless parameters to generate similar results.

Assuming a purely passive neuron (i.e. non-spiking) and is subjected to Gaussian white noise ($\mu = 0$) combined with periodic input, then the membrane potential spectral power distribution reads

$$S(\omega) \sim \frac{1}{1 + 4\pi^2 \tau_m^2 \omega^2} \left( 2\sigma^2 + \frac{A_s^2}{4}(\delta(\omega - \omega_s) + \delta(\omega + \omega_s)) \right), \tag{3}$$

where $\delta(\cdot)$ is the *Dirac* delta function. In other words, the passive LIF-neuron behaves like a low pass filter diminishing larger frequencies and the MTC $\tau_m$ defines the filter's edge frequency $1/\tau_m$.

The total presynaptic input for a neuron $i$ is given by

$$I_{syn}^i = \sum_{j=1}^{N_E} g_{ij}^E S_{ij}(t - t_d)(\nu_i - E_{syn}^j) + \sum_{j=1}^{N_I} g_{ij}^I S_{ij}(t)(\nu_i - E_{syn}^j) \tag{4}$$

where $g_{ij}^{E,\ I}$ are synaptic weights associated with connections between either excitatory (E) and inhibitory (I) pre-synaptic neurons towards a post-synaptic neuron $i$. $E_{syn}$ is the reversal potential for E and I pre-synaptic neurons. The above sum is taken over $N_E$ excitatory and $N_I$ inhibitory pre-synaptic neurons. The resulting synaptic response function $S_{ij}(t)$ at connections from neuron $j$ to neuron $i$ is modeled as

$$S_{ij}(t) = \frac{e^{\frac{t - t_{sp}^j - t_d^{ij}}{\tau_r}} - e^{\frac{t - t_{sp}^j - t_d^{ij}}{\tau_d}}}{\frac{\tau_r}{\tau_d} \left(\frac{\tau_r}{\tau_d - \tau_r}\right) - \frac{\tau_r}{\tau_d} \left(\frac{\tau_d}{\tau_d - \tau_r}\right)} \tag{5}$$

where $t_{sp}^j$ is the time of spike of $j^{th}$ neuron, and $t_d^{ij}$ is the axonal delay between pre-synaptic neuron, $j$, and post-synaptic neuron, $i$. The $\tau_d$ is the decay time constant associated with GABA$_a$ and/or AMPA receptors.

## Spike Timing-Dependent Plasticity (STDP)

Plasticity in our network (both for a pair of neurons and excitatory and inhibitory populations) was modeled using Hebbian spike-timing dependent plasticity [40, 43]. To avoid biased synaptic changes (i.e. preferential LTP/LTD), we chose a symmetric STDP Hebbian learning rule [72, 73]. Specifically, synaptic weight modification in our model follows

$$\Delta g_{\Delta T > 0} = A_+(1 - g/g_{\max})e^{-\Delta T/\gamma^+}$$

$$\Delta g_{\Delta T < 0} = -A_-(g/g_0)e^{\Delta T/\gamma^-} \tag{6}$$

$$g = g + \Delta g$$

Synaptic changes resulting from the rule above are plotted in Fig 3A3. The term $A_+(1 - g/g_{\max})$ and $A_-(g/g_0)$ which depends on the on-line value of synaptic weight, $g(t)$, represent rates of synaptic potentiating and depression respectively, ensure us to be in soft-bound regime [74]. The $\gamma^+$ and $\gamma^-$ are decay time constants. $\Delta T = t_{sp}^{post} - t_{sp}^{pre}$ represents the time difference between the spiking time of pre- and post-synaptic neurons. Whenever $\Delta T$ is positive (negative) the synaptic weight between *pre* to *post* neurons gets potentiated (depressed). The constant $g_{\max}$ denotes the maximum achievable synaptic weight, while $g_0$ denotes the initial synaptic weight, uniform across all synaptic connections prior to learning.

Synaptic changes are bounded within a physiologically relevant range by setting the maximum and minimum value of synaptic weight to $g_{min} = 0.01g_0$ and $g_{max} = 2g_0$. Therefore, every

time the synaptic weights overpass these limits, this condition will impose the value to those mentioned limits to ensure the synaptic weight remains between boundaries. Baseline synaptic connectivity and threshold were selected to set the network in a weak coupling, subthreshold regime, in which an isolated pre-synaptic spike does not guarantee post-synaptic firing. Throughout this report, we used Eq 6 for synaptic modification and our choice of STDP parameters are: $A_+ = 2A_- = 0.02$ and $\gamma_\pm = 10\ ms$.

In Fig 3A3 The effective synaptic weight in coupled neurons is estimated based on the frequency distribution of spiking time differences $\Delta T = t_{sp}^{(2)} - t_{sp}^{(1)}$ between post and pre-synaptic neurons (such as Fig 3A1 and 3A2). As explained above, the STDP rule, Eq 6, defines the synaptic weight change subjected to $\Delta T$. Then the mean synaptic weight change is given by the STDP-rule weighted by the frequency of $\Delta T$

$$\overline{\Delta g} = \int_{-\infty}^{0} A_- e^{\Delta T/\gamma^-} f(\Delta T) d\Delta T + \int_{0}^{\infty} A_+ e^{-\Delta T/\gamma^+} f(\Delta T) d\Delta T \tag{7}$$

$f(\Delta T)$ is the frequency distribution of $\Delta T$. Note that the distribution varies by changing the stimulation frequency. The outcome of this integral predicts the direction of synaptic modification at each stimulation frequency (cf. Fig 3A3).

## Intra- and inter-laminar network model

We modeled a sparse network of leaky-integrate-and-fire (LIF) neurons (see Eq 1), with a 4:1 ratio of excitatory and inhibitory neurons [75] and with fixed intra-laminar connection probability of 0.1 [50, 76]. The choice of LIF neurons is motivated by the need to balance physiological relevance and computational tractability for the network sizes we considered (see below) [77]. The baseline synaptic weights and other parameters have been selected within the reported physiological range [50] and in line with previous studies on LIF cortical network models (see [40, 71, 77, 78] and references therein), and are further summarized in Table 2. In the intra-laminar case, $N = 10000$ (i.e. 8000 excitatory cells, 2000 inhibitory cells; Figs 4 and 5), while $N = 12000$ in the inter-laminar case (i.e. 2400 excitatory cells, 600 inhibitory cells per layer; Fig 6. The equal number of neurons in each layer is an arbitrary choice for simulating purposes). To preserve sparse connectivity between cortical layers, the inter-laminar connection probability was set to 0.05 between excitatory neurons of each layer whenever such a connection was observed, and set to zero otherwise [39, 50]. Inter-laminar connections between inhibitory neurons were neglected.

To study the effect of timescale heterogeneity, we randomly sampled neuronal MTCs (i.e. $\tau_m$) from various probability distributions whose statistics are within the experimentally measured range for cortical neurons [27, 29, 32]. In the intra-laminar case (i.e. Figs 4 and 5), $\tau_m$ was randomly sampled from a Gaussian distribution with mean $\mu_{\tau_m} = 10\ ms$ and standard deviation $\sigma_{\tau_m} = 3\ ms$ (see Fig 4A). To quantify synaptic weight modification between neurons with different MTCs, we identified synapses linking presynaptic cells whose MTCs ranged around the mean of this distribution (i.e. $\tau_m^{(0)} \in \{9.5\ 10.5\}\ ms$) towards post-synaptic cells whose MTCs are away from the mean (i.e. $\tau_m^{(0)} \pm 2\ ms$). We index such synapses with $>$ ($<$): $\tau_m^< \in \{\tau_m \leqslant 8\ ms\}$) and ($\tau_m^> \in \{\tau_m \geqslant 12\ ms\}$. We averaged these synaptic weights at every 500ms in the last 5 seconds of stimulation. These were further averaged over 5 independent trials. The resulting synaptic weight distributions are plotted in Fig 5. In Fig 6 to quantify synaptic weight modification, we considered the synapses between neurons that the difference in their MTCs is more than 5ms (i.e. $\tau_m^> - \tau_m^< \geqslant 5\ ms$).

In the inter-laminar case (i.e. Fig 6), we used empirical MTC values measured across cortical layers [27, 29, 32]. The probability density functions for each layer are plotted in Fig 1A. These data were collected from the Allen Institute Cell Feature Database for human neurons [32]. Since the resulting MTCs are distributed over a wider range compared to the intra-laminar case above, we adjusted the neurons' rheobase so that the resulting input current leads the cell in a dynamical regime similar to the one plotted in Fig 4.

## Supporting information

Previous experimental [15, 16] and computational [46] studies have reported no significant increase in firing rate during tACS, in contrast to our simulations where elevated spiking was observed. We have selected a regime of high tACS amplitude to characteririze the influence of MTC heterogeneity on STDP under tCAS. Ought to the fact that STDP operates as a function of spike timing differences (and hence MTC), it is essential to alter the spiking time of pre-and post-synaptic neurons to induce selective synaptic weights modifications. Achievement of this goal, because of the wide range of neurons MTCs, is notably possible through increasing the amplitude of stimulation, resulting in increase in firing rates. We have nonetheless extended our analyses to low tACS amplitude regimes to evaluate the whether MTC-specific synaptic changes may occur in absence of significant changes in firing rates during tACS at various frequencies. We first revisited our simulations for smaller tACS amplitudes ($A_s = 0.2 \ mV \approx 1 \ mA \approx 1 \ V/m$, See [53]), [15, 16] over an extended time ($t = 5 - 110 \ s$). As shown in S1 Fig, weak stimulation amplitude is capable of entraining neuron spike timing while preserving firing rates. Small, yet noticeable synaptic modification can nonetheless be observed. Compared to results presented in Fig 4 (high tACS amplitude regime) this occurs through the expense of losing the specificity in spiking time/phases, which translates into a reduction of STDP-mediated synaptic modifications. We next considered parameters that would support increased sensitivity to tACS at smaller amplitudes, such as distance to threshold. Modelling this through an additional input current ($\mu_{new} = \mu + \Delta\mu = 5.7 \ mV$ in $I_\zeta$, Eq 1) does change firing rates while modifying the aforementioned specified synapses (see S2 Fig). We also confirmed that no significant synaptic changes would be induced in these regimes even over long stimulation periods. These additional simulations show that stimulation-induced changes in synaptic weights scale with tACS amplitude. Despite no significant change in firing rate, spike timing phase-locking lead to small MTC-specific synaptic modifications (cf. S1 and S2 Figs). These observations are in line with results previously reported [15, 16, 46].

**S1 Fig. Network activity in low stimulation amplitude regime.** (A1) to (A3) are showing the distribution of neurons' firing rate at stimulation frequencies $\omega_s = 20, 25, 30$Hz, respectively, at different time points: ($t = 1, 2s$) pre-stimulation epoch, ($t = 50, 100s$) stimulation epoch, and ($t = 116, 119s$) post-stimulation epoch. (B1) to (B3) are the raster plots of neural population's spiking activity, at mentioned time points and stimulation frequencies. (C1) to (C3) show the distribution of synaptic weights at stimulation frequencies 20, 25, and 30 Hz, and among different sets of synapses, $g_{E \to E}$, $g_{E \to I}$, and $g_{I \to E}$. The stimulation amplitude is $A_s = 0.2mV$. (EPS)

**S2 Fig. Network activity in low stimulation amplitude regime and smaller distance to threshold.** The distance to threshold of all neurons, is reduced by increasing the input current to $\mu = 5.7mV$. (A1) to (A3) are showing the distribution of neurons' firing rate at stimulation frequencies $\omega_s = 20, 25, 30$Hz, respectively, at different time points: ($t = 1, 2s$) pre-stimulation epoch, ($t = 50, 100s$) stimulation epoch, and ($t = 116, 119s$) post-stimulation epoch. (B1) to (B3) are the raster plots of neural population's spiking activity, at mentioned time points and stimulation frequencies. (C1) to (C3) show the distribution of synaptic weights at stimulation

frequencies 20, 25, and 30 Hz, and among different sets of synapses, $g_{E \to E}$, $g_{E \to I}$, and $g_{I \to E}$. The stimulation amplitude is $A_s = 0.2mV$.
(EPS)

## Acknowledgments

AP would like to thank Dr. Farhad Daei for his contribution in developing the simulated code.

## Author Contributions

**Conceptualization:** Aref Pariz, Jeremie Lefebvre.

**Data curation:** Aref Pariz, Daniel Trotter.

**Formal analysis:** Aref Pariz, Daniel Trotter, Axel Hutt, Jeremie Lefebvre.

**Funding acquisition:** Jeremie Lefebvre.

**Investigation:** Aref Pariz.

**Methodology:** Aref Pariz, Jeremie Lefebvre.

**Supervision:** Jeremie Lefebvre.

**Validation:** Aref Pariz, Daniel Trotter, Axel Hutt, Jeremie Lefebvre.

**Visualization:** Aref Pariz.

**Writing – original draft:** Aref Pariz.

**Writing – review & editing:** Aref Pariz, Daniel Trotter, Axel Hutt, Jeremie Lefebvre.

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
