## [Decision Letter · Decision Letter 0]

30 Jan 2023

Dear Dr Pariz,

Thank you very much for submitting your manuscript "Selective control of synaptic plasticity in heterogeneous networks through transcranial alternating current stimulation" for consideration at PLOS Computational Biology.

As with all papers reviewed by the journal, your manuscript was reviewed by members of the editorial board and by several independent reviewers. In light of the reviews (below this email), we would like to invite the resubmission of a significantly-revised version that takes into account the reviewers' comments. In particular, the authors must ensure that the relationship between all features of their simulations and the experimental literature on tACS is well described - both where their simulations concur with the data, and where they diverge. 

Furthermore, please consider using a different colormap in figure 3, one which is sequential and perceptually uniform.

We cannot make any decision about publication until we have seen the revised manuscript and your response to the reviewers' comments. Your revised manuscript is also likely to be sent to reviewers for further evaluation.

Sincerely,

Daniel Bush

Academic Editor

PLOS Computational Biology

Daniele Marinazzo

Section Editor

PLOS Computational Biology

Reviewer's Responses to Questions

**Comments to the Authors:**

Reviewer #1: This manuscript describes a really quite elegant series of simulations investigating the effect of timescale heterogeneity on synaptic plasticity during tACS. The simulations are based on Leaky-Integrate-and-Fire (LIF) excitatory and inhibitory model neurons to develop two neuron network motifs, intra- and inter- laminar cortical circuit models. Neurons and/or networks are assumed to be in asynchronous spiking state to quantify tACS-induced synaptic changes in the absence of endogenous oscillations. Plasticity in the network is modelled using Hebbian spike-timing dependent plasticity (STDP). The results demonstrate that time scale variations through membrane time constant heterogeneity enables tACS to guide inter-neuronal, intra-laminar and inter-laminar synaptic plasticity in a directional and frequency-specific manner. Stimulation using tACS leads to entrainment and the consequent directional synaptic weight modification such that the synapses between average and fast neutrons undergo LTD, while those between average and slow neutrons undergo LTP, irrespective of cell type. The magnitude and direction of overall synaptic weights is stimulation frequency-dependent. Their model has certain limitations, including but not limited to simplified cortical connectivity, unaccounted cell-specific variations and weak connectivity regime. Nonetheless, this interdisciplinary framework makes a strong case for the timescale heterogeneity as a means of guiding synaptic plasticity using non-invasive stimulation paradigm (tACS).

Computational code used for this study and their data are not publicly available. Please ensure it is available without any restriction.

The manuscript is well written.

Minor comments:

Results- page 3- line 92 - Please consider expanding the sentence rather than putting the second condition in brackets, to ensure a smooth flow.

Neurons with smaller (larger)....shorter (longer)...

Consider replacing it with ‘neurons with smaller MTC exhibit shorter integration times and those with larger MTC exhibit longer integration times’, or other format that you might like.

Fig 3 legend: The top (bottom)...

‘The top and bottom circles denote the MTC of neuron(2) and neuron (1), respectively.’

Please stay consistent. At some places, Fig X (Y#) is used, while at other instances, it is Fig X Y#

Discussion: Page 10, line 338 - Please modify the sentence: ‘as well as a other…, ought to play’.

Reviewer #2: Pariz et al. build heterogeneous neural network models to explore the effects of transcranial alternating current stimulation (tACS) on neural behavior. This is an interesting and sufficiently novel research goal, however, there are several principal weaknesses. The network models are not clearly described, I do see any constrains to their parameters, and they demonstrate behavior that is known to not exist in the real neural networks during tACS. Thus, interpretability of the results is not high.

First of all, I am concerned about the properties of neural networks. From Figure 4E it is evident that the spiking rate increased manifolds during stimulation. That is contrary to the expected behavior - all experimental neural studies (Johnson et al., Science Adv, 2019; Krause et al., PNAS, 2019), as well as computational models (Tran et al., Neuroimage, 2022; Obermayer and colleagues in PLOS Comp. Biol.), showed a little-to-no increase in spiking rate during tACS. Thus, the present models do not capture the critical features of their biological counterparts.

The description of the computational network models is confusing. Figure 1 depicts realistic neurons, and some parts of the paper give the impression that two networks of realistic neurons were simulated. However, other methods described radically more simple leaky integrate-and-fire (LIF) neurons. The authors should avoid misrepresentation.

Finally, the authors provided no constraints for the computational parameters in their network models.

**Have the authors made all data and (if applicable) computational code underlying the findings in their manuscript fully available?**

Reviewer #1: **No: **Git repositories are not public.

Reviewer #2: **No: **The github link that authors provided is empty.

PLOS authors have the option to publish the peer review history of their article (what does this mean?). If published, this will include your full peer review and any attached files.

Reviewer #1: No

Reviewer #2: No
---

## [Decision Letter · Decision Letter 1]

28 Mar 2023

Dear Dr Pariz,

We are pleased to inform you that your manuscript 'Selective control of synaptic plasticity in heterogeneous networks through transcranial alternating current stimulation (tACS)' has been provisionally accepted for publication in PLOS Computational Biology.

Best regards,

Daniel Bush

Academic Editor

PLOS Computational Biology

Daniele Marinazzo

Section Editor

PLOS Computational Biology

Reviewer's Responses to Questions

**Comments to the Authors:**

Reviewer #1: The authors have diligently addressed all the comments.

Reviewer #2: The authors addressed all issues.

**Have the authors made all data and (if applicable) computational code underlying the findings in their manuscript fully available?**

Reviewer #1: Yes

Reviewer #2: Yes

PLOS authors have the option to publish the peer review history of their article (what does this mean?). If published, this will include your full peer review and any attached files.

Reviewer #1: No

Reviewer #2: No

---

## [Editor Report · Acceptance letter]

20 Apr 2023

PCOMPBIOL-D-22-01659R1 

Selective control of synaptic plasticity in heterogeneous networks through transcranial alternating current stimulation (tACS)

Dear Dr Pariz,

I am pleased to inform you that your manuscript has been formally accepted for publication in PLOS Computational Biology. Your manuscript is now with our production department and you will be notified of the publication date in due course.

With kind regards,

Anita Estes
